# Kernel-Free Quadratic Surface Regression for Multi-Class Classification

**DOI:** 10.3390/e25071103

**Published:** 2023-07-24

**Authors:** Changlin Wang, Zhixia Yang, Junyou Ye, Xue Yang

**Affiliations:** 1College of Mathematics and Systems Science, Xinjiang University, Urumuqi 830046, China; 2Institute of Mathematics and Physics, Xinjiang University, Urumuqi 830046, China

**Keywords:** multi-class classification, least squares regression, quadratic surface, kernel-free trick, ε-dragging technique

## Abstract

For multi-class classification problems, a new kernel-free nonlinear classifier is presented, called the hard quadratic surface least squares regression (HQSLSR). It combines the benefits of the least squares loss function and quadratic kernel-free trick. The optimization problem of HQSLSR is convex and unconstrained, making it easy to solve. Further, to improve the generalization ability of HQSLSR, a softened version (SQSLSR) is proposed by introducing an ε-dragging technique, which can enlarge the between-class distance. The optimization problem of SQSLSR is solved by designing an alteration iteration algorithm. The convergence, interpretability and computational complexity of our methods are addressed in a theoretical analysis. The visualization results on five artificial datasets demonstrate that the obtained regression function in each category has geometric diversity and the advantage of the ε-dragging technique. Furthermore, experimental results on benchmark datasets show that our methods perform comparably to some state-of-the-art classifiers.

## 1. Introduction

Consider a training set:(1)$$\begin{array}{c} T_{1}={\left\{\left(x_{i},y_{i}\right)\right\}}_{i=1}^{n}, \end{array}$$
comprising *n* samples, each represented by a *d*-dimensional vector $x_{i}\in R^{d}$, and a corresponding label $y_{i}\in \left\{1,2,\cdots ,K\right\}$, indicating the class of sample in *K* classes.

For multi-class classification, one popular strategy is to encode each label using one-hot encoding. Consequently, the original training set: $T_{1}$ (Equation 1) is transformed into a new training set
(2)$$\begin{array}{c} T_{2}={\left\{\left(x_{i},y_{i}\right)\right\}}_{i=1}^{n}, \end{array}$$
where each sample corresponds to a label vector $y_{i}=\mathrm{one}-\mathrm{hot}\left(y_{i}\right)$ (Definition 3). Our goal is to find *K* functions $f_{k}\left(x\right),k=1,2,\ldots ,K$ that satisfy $f\left(x_{i}\right)\approx y_{i}$, where $f\left(x_{i}\right)={\left(f_{1}\left(x_{i}\right),f_{2}\left(x_{i}\right),\cdots ,f_{K}\left(x_{i}\right)\right)}^{T}$ for i=1,2,⋯,n. Once these *K* functions are determined, a new sample x can be classified using the decision rule
(3)$$\begin{array}{c} g\left(x\right)=\underset{k=1,2,\cdots K}{\arg\max}f_{k}\left(x\right). \end{array}$$

In recent years, numerous studies have focused on the multi-class classification problem. In 1994, Imran Naseem et al. [1,2] proposed the original least square regression classifier (LSR) based on the label vectors. This method assigns input samples to the class represented by the label vector closest to the predicted vector. To improve the accuracy of LSR, Xian et al. [3] introduced the ε-dragging technique to expand the interval between different classes, creating a discriminative LSR (DLSR). Zhang et al. [4] proposed a retargeted LSR (ReLSR) which learns soft labels with large margin constraints directly from training data. Wen et al. [5] proposed an inter-class sparsity DLSR (ICS_DLSR) by introducing inter-class sparsity constraints. Wang et al. [6] proposed a relaxed group low-rank regression model (RGLRR) that incorporates sparsity consistency and graph embedding into the group low-rank regression model. Recently, scholars have proposed several methods to improve the classification accuracy of DLSR, including the margin scalable DLSR (MSDLSR) [7], the robust DLSR (RODLSR) [8], regularized label relaxation linear regression (RLRLR) [9], low-rank DLSR (LRDLSR) [10], and discriminative least squares regression based on within-class scatter minimization (WSCDLSR) [11]. To improve the classification accuracy of ReLSR, Zhang et al. [12] introduced the intra-class compactness graph into ReLSR, proposing the discriminative marginalized LSR (DMLSR). Additionally, LSR has been extended for feature selection by Zhang et al. [13] and Zhao et al. [14]. All of the above methods are linear classification models, which have less computation time but have difficulty handling nonlinearly separable data. The kernel ridge regression classifier (KRR) was proposed to address the defects previously mentioned, using the kernel trick [15,16]. However, it is challenging to select the appropriate kernel function and kernel parameter.

In 2008, the quadratic surface SVM (QSSVM) [17] was proposed to address the issue of excessive kernel parameter selection in SVM [18], utilizing a kernel-free technique. Later, Luo et al. [19] introduced the soft margin quadratic SVM (SQSSVC). Subsequently, further studies have been conducted, including classification problems [20,21,22,23], regression problems [24], clustering problems [25], and applications [26,27,28,29].

In this paper, we propose two nonlinear classification models, the hard quadratic surface least squares regression (HQSLSR) and its softened version, the soft quadratic surface least squares regression (SQSLSR). The main contributions of this work are summarized as follows:

(1) We propose a novel nonlinear model (HQSLSR), by utilizing a kernel-free trick, which avoids the difficulty of selecting the appropriate kernel functions and corresponding parameters and maintains good interpretability. Moreover, a softened version (SQSLSR) is developed, which employs the ε-dragging technique to enlarge inter-class distances so that its discriminant ability is improved further.

(2) The proposed HQSLSR yields a convex optimization problem without constraints, which can be directly solved. An alteration iteration algorithm is designed for SQLSR, which involves only the convex optimization problem and leads to quick convergence. Additionally, the computational complexity and interpretability of our methods are also discussed.

(3) In numerical experiments, the geometric intuition and advantage of the ε-dragging technique for our methods on artificial datasets are demonstrated. The experimental results over benchmark datasets exhibit that our methods achieve comparable accuracy to other nonlinear classifiers while requiring less computational time cost.

This paper is organized as follows. Section 2 briefly describes related work. Section 3 presents the proposed HQSLSR and SQSLSR models and their respective algorithms. Section 4 discusses relevant characteristics. Section 5 presents experimental results, and finally, we conclude in Section 6.

## 2. Related Works

In this section, following the presentation of notations, we provide a concise introduction to two fundamental approaches: least squares regression classifiers (LSR) [1] and discriminative least squares regression classifiers (DLSR) [3].

### 2.1. Notations

We begin by presenting the notations employed in this paper. Lowercase boldface and uppercase boldface fonts represent vectors and matrices, respectively. The vector ${\left(1,1,\cdots ,1\right)}^{T}\in R^{n}$ is represented by $1_{n}$. Define the zero vector and null matrix as 0 and O, respectively. For a matrix $W={\left(w_{ij}\right)}_{d\times K}$, its *i*-th column is denoted as $w_{i}$. In addition, we give the following three definitions.

**Definition** **1.**
*For any real symmetric matrix $A={\left(a_{ij}\right)}_{d\times d}\in S^{d}$, its half-vectorization operator can be defined as follows:*

$$\begin{array}{c} \mathrm{hvec}\left(A\right)={\left(a_{11},a_{12},\cdots ,a_{1d},a_{22},\cdots ,a_{2d},\cdots ,a_{dd}\right)}^{T}\in R^{\frac{d^{2}+d}{2}}. \end{array}$$



**Definition** **2.**
*For any vector $x={\left(x_{1},x_{2},\cdots ,x_{d}\right)}^{T}$, its quadratic vector with cross terms can be defined as follows:*

$$\begin{array}{c} \mathrm{lvec}\left(x\right)={\left(\frac{1}{2}x_{1}^{2},x_{1}x_{2},\cdots ,x_{1}x_{d},\frac{1}{2}x_{2}^{2},x_{2}x_{3},\cdots ,\frac{1}{2}x_{d}^{2}\right)}^{T}\in R^{\frac{d^{2}+d}{2}}. \end{array}$$



**Definition** **3.**
*For any given positive integer k∈{1,2,⋯,K}, the one-hot encoding operator is defined as follows:*

$$\begin{array}{c} \mathrm{one}-\mathrm{hot}\left(k\right)=e_{k}, \end{array}$$

*where $e_{k}$ is the K-dimensional unit vector, with the k-th element 1.*


### 2.2. Least Squares Regression Classifier

Given a training set $T_{2}$ (Equation 2), the goal of LSR is to find the following *K* linear functions:(4)$$\begin{array}{c} f_{k}\left(x\right)=w_{k}^{T}x+c_{k},\;k=1,2,\cdots ,K, \end{array}$$
where $w_{k}\in R^{d},\;c_{k}\in R,\;k=1,2,\cdots ,K$.

To obtain the *K* linear functions (Equation 4), the following optimization problem is formulated as
(5)$$\begin{array}{c} \min_{W,c}∥X^{T}W+1_{n}c^{T}{-Y∥}_{F}^{2}+\lambda {∥W∥}_{F}^{2}, \end{array}$$
where the sample matrix $X=\left(x_{1},x_{2},\cdots ,x_{n}\right)\in R^{d\times n}$ is formed by all the samples in the training set $T_{2}$ (Equation 2), the label matrix $Y={\left(y_{1},y_{2},\cdots ,y_{n}\right)}^{T}\in R^{n\times K}$ is formed by the label vectors in $T_{2}$ (Equation 2), and $W=\;\left(w_{1},w_{2},\cdots ,w_{K}\right)\in R^{d\times K}$, $c=\;{\left(c_{1},c_{2},\cdots c_{K}\right)}^{T}\;\in R^{K}$ are formed by the normal vectors and biases of the *K* linear functions (Equation 4), respectively.

Clearly, the optimization problem (Equation 5) is a convex optimization problem, and its solution has the following form:$$\begin{array}{c} W={\left(XHX^{T}\;+\;\lambda I\right)}^{-1}XHY,\;c=\frac{1}{n}\left(Y^{T}1_{n}-W^{T}X1_{n}\right), \end{array}$$
where $H=I-\frac{1}{n}1_{n}1_{n}^{T}$. Thus, once the solutions W, c of the optimization problem (Equation 5) is obtained, we can find the *K* linear functions.

For a new sample $x\in R^{d}$, its class is obtained by the following decision function:(6)$$\begin{array}{c} g\left(x\right)=\underset{k=1,2,\cdots K}{\arg\max}w_{k}^{T}x+c_{k}. \end{array}$$

### 2.3. Discriminative Least Squares Regression Classifier

Xiang et al. [3] proposed the discriminative least squares regression classifier (DLSR) to improve the classification performance of LSR.

For the training set $T_{2}$ (Equation 2), we define the constant matrix $B={\left(B_{ik}\right)}_{n\times K}$ as follows:(7)$$B_{ik}=\left\{\begin{array}{cc} +1, & \mathrm{if}\;y_{ik}=1,\\ -1, & \mathrm{otherwise}, \end{array}\right.$$
where $y_{ik}$ represents the *k*-th component of the label vector $y_{i}$ of the *i*-th sample, the optimization problem of DLSR is formulated as follows:(8)$$\begin{array}{cc} \min_{W,c,E} & {∥X^{T}W+1_{n}c^{T}-Y-B⊙E∥}_{F}^{2}+\lambda {∥W∥}_{F}^{2},\\ s.t.\;\; & E\geq O, \end{array}$$
where ⊙ is the Hadamard product of matrices. $E={\left(\varepsilon_{ik}\right)}_{n\times K}$ is an ε-dragging matrix to be found, and each of its non-negative elements $\varepsilon_{ik}$ is called the ε-dragging factor.

It is evident that DLSR takes into account the inter-class distance based on LSR. Specifically, DLSR increases inter-class distances by introducing the ε-dragging technique, causing different classes of regression targets to move in opposite directions.

## 3. Kernel-Free Nonlinear Least Squares Regression Classifiers

For multi-class classification problems with the training set $T_{2}$ (Equation 2), we propose the hard quadratic surface least squares regression classifier (HQSLSR) and its softened version (SQSLSR). The relevant properties of our methods are also analyzed theoretically.

### 3.1. Hard Quadratic Surface Least Squares Regression Classifier

For the training set $T_{2}$ (Equation 2), we aim to find *K* quadratic functions as follows:(9)$$\begin{array}{c} f_{k}\left(x\right)=\frac{1}{2}x^{T}A_{k}x+b_{k}^{T}x+c_{k},\;k=1,2,\cdots ,K, \end{array}$$
where $A_{k}\in S^{d},B_{k}\in R^{d}$, $c_{k}\in R$. If these *K* quadratic functions are found, the label of a new sample x is determined by the following decision rule:(10)$$\begin{array}{c} g\left(x\right)=\underset{k=1,2,\cdots K}{\arg\max}\frac{1}{2}x^{T}A_{k}x+b_{k}^{T}x+c_{k}. \end{array}$$

In order to find the *K* quadratic functions (Equation 9), we construct the following optimization problem:(11)$$\begin{array}{c} \min_{A_{k},b_{k},c_{k}}\sum_{i=1}^{n}\sum_{k=1}^{K}{\left(\frac{1}{2}x_{i}^{T}A_{k}x_{i}+b_{k}^{T}x_{i}+c_{k}-y_{ik}\right)}^{2}+\lambda \sum_{k=1}^{K}({∥\mathrm{hvec}\left(A_{k}\right)∥}_{2}^{2}+{∥b_{k}∥}_{2}^{2}), \end{array}$$
where λ is the regularization parameter, $\mathrm{hvec}(A_{k})$ is a vector by Definition 1, which is constituted by the upper triangular elements of the symmetry matrix $A_{k}$, and $y_{ik}$ indicates the *k*-th component of the label vector $y_{i}$ of the *i*-th sample. For the objective function (Equation 11), its first term minimizes the sum of the squares of the errors between the real and predicted label; the second term is a regularization term about the model coefficients, which aims to enhance the generalization ability of our model. It is worth noting that the upper triangular elements of the matrix $A_{k}$ instead of all elements are involved in the regularization term by using the symmetry of the matrix.

For convenience, by using the symmetry of the matrix $A_{k}$ and following Definitions 1 to 2, the first term of the objective function in the optimization problem (Equation 11) is simplified as follows:(12)$$\begin{array}{cc} \; & \sum_{i=1}^{n}\sum_{k=1}^{K}{\left(\frac{1}{2}x_{i}^{T}A_{k}x_{i}+b_{k}^{T}x_{i}+c_{k}-y_{ik}\right)}^{2}\\ = & \sum_{i=1}^{n}\sum_{k=1}^{K}{\left(\mathrm{hvec}{\left(A_{k}\right)}^{T}\mathrm{lvec}\left(x_{i}\right)+b_{k}^{T}x_{i}+c_{k}-y_{ik}\right)}^{2}\\ = & \sum_{i=1}^{n}\sum_{k=1}^{K}{\left(w_{k}^{T}z_{i}+c_{k}-y_{ik}\right)}^{2}, \end{array}$$
where
(13)$$\begin{array}{cc} \; & w_{k}={\left(\mathrm{hvec}{\left(A_{k}\right)}^{T},b_{k}^{T}\right)}^{T},\;k=1,\cdots ,K, \end{array}$$
(14)$$\begin{array}{cc} \; & z_{i}={\left(\mathrm{lvec}{\left(x_{i}\right)}^{T},x_{i}^{T}\right)}^{T},\;i=1,\cdots ,n. \end{array}$$ By Equation (Equation 13), minimizing $\sum_{k=1}^{K}(∥\mathrm{hvec}\left(A_{k}\right){∥}_{2}^{2}+∥b_{k}{∥}_{2}^{2})$ is equivalent to minimizing $\sum_{k=1}^{K}{∥w_{k}∥}_{2}^{2}$. Furthermore, combining Equation (Equation 12), the optimization problem (Equation 11) is further formulated as
(15)$$\begin{array}{c} \min_{W,c}J_{1}\left(W,c\right)=\|Z^{T}W+1_{n}c^{T}{-Y\|}_{F}^{2}+\lambda {\left\|W\right\|}_{F}^{2}, \end{array}$$
where $Z=\left(z_{1},z_{2},\cdots ,z_{n}\right)\in R^{\frac{d^{2}+3d}{2}\times n},W=\left(w_{1},\;w_{2},\cdots ,w_{K}\right)\in R^{\frac{d^{2}+3d}{2}\times K},$ $c\;=\;{\left(c_{1},c_{2},\cdots ,c_{K}\right)}^{T}$
$\in R^{K}$.

Next, the solution of the optimization problem (Equation 15) is given by the following theorem.

**Theorem** **1.***The optimal solution of the optimization problem *(Equation 15)* is as follows*(16)$$\begin{array}{cc} \; & W={\left(ZHZ^{T}+\lambda I\right)}^{-1}ZHY, \end{array}$$(17)$$\begin{array}{cc} \; & c=\frac{1}{n}\left(Y^{T}1_{n}-W^{T}Z1_{n}\right), \end{array}$$
where $H=I-\frac{1}{n}1_{n}1_{n}^{T}$.

**Proof.** Obviously, Formula (Equation 15) is a convex optimization problem. According to the optimality condition of the unconstrained optimization problem, we have
(18)$$\begin{array}{cc} \; & \nabla_{c}J_{1}\left(W,c\right)=W^{T}Z1_{n}\;+\;c1_{n}^{T}1_{n}\;-\;Y^{T}1_{n}=0, \end{array}$$
(19)$$\begin{array}{cc} \; & \nabla_{W}J_{1}\left(W,c\right)\;=ZZ^{T}W\;+\;Z1_{n}c^{T}\;-\;ZY\;+\;\lambda W=0. \end{array}$$According to Equation (Equation 18), we obtain
(20)$$\begin{array}{c} c=\frac{1}{n}\left(Y^{T}1_{n}\;-W^{T}Z1_{n}\right). \end{array}$$By substituting Equation (Equation 20) into Equation (Equation 19), we have
(21)$$\begin{array}{c} W={\left(ZHZ^{T}+\lambda I\right)}^{-1}ZHY, \end{array}$$
where $H=I-\frac{1}{n}1_{n}1_{n}^{T}$.    □

After solving the optimization problem (Equation 15) from Theorem 1, $W_{k}$ and $c_{k}$ are obtained by the *k*-th column of matrix w and the *k*-th component of vector c, respectively. Then, $A_{k}$ and $b_{k}$ can be obtained by Equation (Equation 13). Therefore, the decision function in Equation (Equation 10) can be established.

### 3.2. Soft Quadratic Surface Least Squares Regression Classifier

In this subsection, we propose the SQSLSR by introducing the ε-dragging factor into the HQSLSR. For the training set $T_{2}$ (Equation 2), the following optimization problem is constructed:(22)$$\begin{array}{cc} \min_{} & \;\;\sum_{i=1}^{n}\sum_{k=1}^{K}{\left(\frac{1}{2}x_{i}^{T}A_{k}x_{i}+\;b_{k}^{T}x_{i}\;+\;c_{k}\;-\;\left(y_{ik}\;+\;B_{ik}\varepsilon_{ik}\right)\right)}^{2}\;\;+\lambda \;\;\sum_{k=1}^{K}(∥\mathrm{hvec}\left(A_{k}\right){∥}_{2}^{2}+∥b_{k}{∥}_{2}^{2}),\\ s.t. & \;\varepsilon_{ik}\geq 0,\;\;i=1,2,\cdots ,n,\;\;\;k=1,2,\cdots ,K, \end{array}$$
where $A_{k},b_{k},c_{k},\varepsilon_{ik},\;i=1,2,\cdots ,n,\;\;\;k=1,2,\cdots ,K$ are variables to be found, respectively. $\varepsilon_{ik}\geq 0$ is the ε-dragging factor, and the constant $B_{ik}$ is defined in detail in Equation (Equation 7). The distance between the label vectors of different classes is expanded by using the ε-dragging factor. Therefore, compared with the HQSLSR model, the SQSLSR model distinguishes samples from different classes more easily.

For simplicity, by defining the ε-dragging matrix E as being similar to the transformation of the optimization problem (Equation 11), the optimization problem (Equation 22) is equivalently expressed as follows:(23)$$\begin{array}{cc} \min_{W,c,E} & J_{2}\left(W,c,E\right)=\|Z^{T}W+1_{n}c^{T}{-\left(Y+B⊙E\right)\|}_{F}^{2}+\lambda {\left\|W\right\|}_{F}^{2},\\ s.t. & \;E\geq O, \end{array}$$
where E≥O means that the elements of the matrix E are non-negative. To solve the optimization problem (Equation 23), we use the alternating iteration method.

First, update W and c. By fixing the dragging matrix E and letting $\tilde{Y}=Y+B⊙E$, the optimization problem (Equation 23) is simplified as follows:(24)$$\begin{array}{c} \min_{W,c}\|Z^{T}W+1_{n}c^{T}-\tilde{Y}{\|}_{F}^{2}+\lambda {\left\|W\right\|}_{F}^{2}. \end{array}$$

Similar to the solution of the optimization problem (Equation 15), the iterative equation for the optimization problem (Equation 24) with respect to W and c is as follows: (25)$$\begin{array}{cc} \; & W={\left(ZHZ^{T}+\lambda I\right)}^{-1}ZH\tilde{Y}, \end{array}$$(26)$$\begin{array}{cc} \; & c=\frac{1}{n}\left({\tilde{Y}}^{T}1_{n}-W^{T}Z1_{n}\right), \end{array}$$
where $H=I-\frac{1}{n}1_{n}1_{n}^{T}$.

Then, update the draggings matrix E. By fixing W, c and letting the residual matrix $R=Z^{T}W+1_{n}c^{T}-Y$, the optimization problem (Equation 23) is transformed into
(27)$$\begin{array}{cc} \min_{E} & {\left\|R-B⊙E\right\|}_{F}^{2},\\ s.t. & \;E\geq O. \end{array}$$ The solution to the optimization problem (Equation 27) can be obtained by the following equation:(28)$$\begin{array}{c} E=\max(B⊙R,O). \end{array}$$

Specifically, according to the definition of the Frobenius norm, solving the optimization problem (Equation 27) is equivalent to solving the following n×K subproblems:(29)$$\begin{array}{cc} \min_{\varepsilon_{ik}} & {\left(R_{ik}-B_{ik}\varepsilon_{ik}\right)}^{2},\\ s.t. & \;\varepsilon_{ik}\geq 0,\;\;\;\;i=1,2,\ldots ,n,\;\;k=1,2,\ldots ,K, \end{array}$$
where $R_{ik}$ is the element of the *i*-th row and *k*-th column of the matrix R. Since $B_{ik}^{2}=1$, we have ${\left(R_{ik}-B_{ik}\varepsilon_{ik}\right)}^{2}={\left(B_{ik}R_{ik}-\varepsilon_{ik}\right)}^{2}$. Then the solution to the optimization problem (Equation 29) is $\varepsilon_{ik}=\max\left(B_{ik}R_{ik},0\right)$. Thus, Equation (Equation 28) is the solution to the optimization problem (Equation 27).

Through the above solution process, we briefly summarize the algorithm of the optimization problem (Equation 23) as follows:

After obtaining $A_{k},b_{k},c_{k},\;k=1,2,\ldots ,K$ by Algorithm 1, the corresponding decision function (Equation 10) can also be constructed.

**Algorithm 1** SQSLSR**Input:** Training set $T_{2}=\left\{\left(x_{i},y_{i}\right)|x_{i}\in R^{d},y_{i}\in R^{K}\right\}$, maximum iteration number T=20, parameter λ1:Define the matrix E, $W,W_{0}$  and vector c, $c_{0}$2:Initialize E=O, $W_{0}=O$, $c_{0}=0$3:Transform $Z_{i}\;\;i=1,2,\ldots ,n,$ by (Equation 14)4:Construct the matrix $Z=(z_{1},z_{2},\cdots ,z_{n})$ and $Y={\left(y_{1},y_{2},\cdots ,y_{n}\right)}^{T}$5:Calculate $H=I-\frac{1}{n}1_{n}1_{n}^{T}$ and $V={\left(ZHZ^{T}+\lambda I\right)}^{-1}ZH$6:**for** t=1:T **do**7:   $\tilde{Y}=Y+B⊙E$8:   Calculate $W=V\tilde{Y}$9:   Calculate c by (Equation 26)10:   Calculate E by (Equation 28)11:   **if** $\|W-W_{0}{\|}_{F}^{2}+{\left\|c-c_{0}\right\|}_{2}^{2}\leq 10^{-3}$ **then**12:     stop13:   **end if**14:   $W_{0}=W$, $\;c_{0}=c$15:**end for**16:Calculate $A_{k}$, $b_{k}$ and $c_{k}$ by the inverse operation of $w_{k}=(\mathrm{hvec}{\left(A_{k}\right)}^{T},$
$b_{k}^{T}{)}^{T}$, where k=1,2,⋯,K, $W_{0}=\left(w_{1},\;w_{2},\cdots ,w_{K}\right)$, and $c_{0}={\left(c_{1},c_{2},\cdots ,c_{K}\right)}^{T}$
**Output:** $A_{k},b_{k},c_{k},\;k=1,2,\ldots ,K$.

## 4. Discussion

In this section, we first discuss the convergence of Algorithm 1. Then, we discuss the computational complexities of HQSLSR and SQSLSR, respectively. Lastly, we analyze their interpretability.

### 4.1. Convergence Analysis

Since Algorithm 1 adopts an iterative method to solve the optimization problem (Equation 23), its convergence is discussed in this subsection.

**Theorem** **2.**
*If the sequence of iterations $\left\{W^{t},c^{t},E^{t}\right\}$ can be obtained by Algorithm 1, then the objective function $J_{2}\left(W^{t},c^{t},E^{t}\right)$ of the optimization problem (Equation 23) is monotonically decreasing.*


**Proof.** First, let *t* be the number of current iterations. Then, we define the value of the objective function of the optimization problem (Equation 23) as $J_{2}\left(W^{t},c^{t},E^{t}\right)$.By the strong convexity of the optimization problem, given $E^{t}$, $W^{t+1}$ and $c^{t+1}$ can be obtained from Equations (Equation 25) and (Equation 26), respectively, and have the following inequality:
(30)$$J_{2}\left(W^{t+1},c^{t+1},E^{t}\right)\leq J_{2}\left(W^{t},c^{t},E^{t}\right).$$Then, fixing $W^{t+1}$ and $c^{t+1}$, $E^{t+1}$ can be obtained from Equation (Equation 28), and with the following inequality:
(31)$$\begin{array}{c} J_{2}\left(W^{t+1},c^{t+1},E^{t+1}\right)\leq J_{2}\left(W^{t+1},c^{t+1},E^{t}\right). \end{array}$$ Combining the inequalities (Equation 30) and (Equation 31), we have the following inequality:
(32)$$\begin{array}{c} J_{2}\left(W^{t+1},c^{t+1},E^{t+1}\right)\leq J_{2}\left(W^{t},c^{t},E^{t}\right), \end{array}$$
Thus, the proof is complete. □

### 4.2. Computational Complexity

In this subsection, we provide a detailed analysis of the computational complexities of our methods. Here, *n*, *d*, and *K* represent the number of samples, features, and classes, respectively. From Definition 1, Definition 2, and Equation (Equation 12), it can be observed that our methods aim to transform the feature dimension of the sample from a *d*-dimensional space to an $l=\frac{d^{2}+3d}{2}$-dimensional space. For simplicity, we ignore the computational cost of addition and subtraction.

The HQSLSR classifier is solved by Equations (Equation 16) and (Equation 17), which involve matrix inversion and multiplication. Therefore, the computational complexity of the HQSLSR classifier is about $O(l^{3}+nl^{2}+\left(n^{2}+nK\right)l)$.

According to Algorithm 1, we briefly analyze the computational complexity of SQSLSR. The computational complexity of SQSLSR is mainly concentrated on steps 5, 8, 9, and 10 of Algorithm 1. Step 5 involves matrix inversion and multiplication, and its computational complexity is $O(l^{3}+nl^{2}+n^{2}l)$. Steps 8, 9, and 10 involve only matrix multiplication, so the computational complexity of each iteration is about O(nKl+nK). In summary, the total computational complexity of SQSLSR is about $O(l^{3}+nl^{2}+n^{2}l+t\left(nKl+nK\right))$, where *t* is the number of iterations.

### 4.3. Interpretability

Although HQSLSR and SQSLSR are kernel-free, they can achieve the goal of nonlinear separation and retain interpretability. Therefore, we further elaborate on their interpretability.

Note that the decision functions of our methods are constructed by the separation quadratic function
(33)$$\begin{array}{c} h\left(x\right)=\frac{1}{2}x^{T}Ax+b^{T}x+c=\frac{1}{2}\sum_{i=1}^{d}\sum_{j=1}^{d}a_{ij}x_{i}x_{j}+\sum_{i=1}^{d}b_{i}x_{i}+c, \end{array}$$
where $x_{i}$ is the *i*-th feature of the vector $x\in R^{d}$, $a_{ij}$ is the element of the *i*-th row and *j*-th column of the symmetry matrix $A\in S^{d}$, and $b_{i}$ is the *i*-th component of the vector $b\in R^{d}$, c∈R. From the quadratic function (Equation 33), we can see that the values of $b_{i}$, $a_{ii}\left(i=j\right),$ and $a_{ij}\;\left(i\neq j\right)$ determine the contributions of the first order term and the second order term of the *i*-th feature $x_{i}$, and the cross term of $x_{i}$ and $x_{j}$, respectively. Roughly speaking, let $\theta_{i,h(x)}=|a_{ii}\left|+\right|a_{ij}\left|+\right|b_{i}|\;\left(j=1,2,\cdots ,d,j\neq i\right)$, the higher the value of $\theta_{i,h(x)}$, the more the *i*-th feature $x_{i}$ contributes to the quadratic function (Equation 33).

For *K* quadratic functions $f_{k}\left(x\right),\;\;k=1,\cdots ,K$ as shown in Equation (Equation 10), let $\theta_{i,k}=\theta_{i,f_{k}\left(x\right)}$ represents the contribution of the *i*-th feature to the *k*-th quadratic function $f_{k}\left(x\right),\;k=1,\cdots ,K$. Let $\theta_{i}=\sum_{k=1}^{K}\theta_{i,k},\;i=1,\cdots d$. The larger $\theta_{i}$ is, the more important the *i*-th feature is to the decision function (Equation 10). In particular, when $\theta_{i}=0$, the *i*-th feature of x does not work. Therefore, our methods have a certain interpretability.

## 5. Numerical Experiments

In this section, we first implement our SQSLSR and HQSLSR on five artificial datasets to show their geometric meaning and compare them with LSR and DLSR. We also carry out our SQSLSR and HQSLSR on 16 UCI benchmark datasets, and compare their accuracy with LSR, DLSR, LRDLSR, WCSDLSR, linear discriminant analysis(LDA), QSSVM, reg-LSDWPTSVM [22], SVM, and KRR. For convenience, SVMs with a linear kernel and rbf kernel are denoted by SVM-L and SVM-R, respectively. KRRs with an RBF kernel and polynomial kernel are denoted as KRR-R and KRR-P, respectively. Remarkably, on multi-class classification datasets, the SVM and QSSVM methods use the one-against-rest strategy [30]. We adopt the five-fold cross-validation to select the parameters in these methods. The regularization parameters of SQSLSR and other methods are selected from the set $\left\{2^{-8},2^{-7},\cdots ,2^{8}\right\}$. The parameters of the RBF kernel and polynomial kernel are selected from the set $\left\{2^{-6},2^{-4},\cdots ,2^{6}\right\}$. All numerical experiments are executed using MATLAB R2020(b) on a computer with a 2.80 GHz (I7-1165G7) CPU and 16 G available memory.

### 5.1. Experimental Results on Artificial Datasets

We construct five artificial datasets to demonstrate the geometric meaning of our methods and the advantage of the ε-dragging technique. Datasets I-IV are binary classifications, where each dataset contains 300 points, and each class has 150 points. Dataset V has three classifications, and each class has 20 points. As the decision functions of our proposed HQSLSR and SQSLSR methods, as well as the comparison methods LSR and DLSR, are all composed of *K* regression functions, we present *K* pairs of regression curves $f_{k}\left(x\right)=\;0\;\mathrm{and}\;\;1,$
k=1,2 to display their classification results. Here, $f_{k}\left(x\right)=1$ is the regression curve of the *k*-th class, $f_{k}\left(x\right)=0$ is the regression curve of samples other than class *k*, *k* = 1, 2.

The first-class samples, $f_{1}\left(x\right)=1$ and $f_{1}\left(x\right)=0$ are indicated by the blue “+”, blue line and blue dotted line, respectively. The second-class samples, $f_{2}\left(x\right)=1$ and $f_{2}\left(x\right)=0$ are represented by the red “∘”, red line and red dotted line, respectively. The accuracy of each method on the artificial dataset is shown in the top right corner.

The artificial dataset I is linearly separable. Figure 1 shows the results of the four methods, including LSR, DLSR, HQSLSR, and SQSLSR. It can be observed that $f_{1}\left(x\right)\;=\;1$ and $f_{2}\left(x\right)\;=\;0$ coincide; $f_{2}\left(x\right)\;=\;1$ and $f_{1}\left(x\right)\;=\;0$ coincide too. The samples of each class come close to the corresponding regression curve, and stay away from the regression curves of the other classes. In addition, the four methods can correctly classify the samples on this linear separable artificial dataset I.

As shown in Figure 2, the artificial dataset II includes some intersecting samples. Our methods outperform LSR and DLSR in terms of classification accuracy, because our HQSLSR and SQSLSR can obtain two pairs of regression curves, while LSR and DLSR can only obtain two pairs of straight regression lines. It is worth noting that the accuracy of SQSLSR is slightly higher than that of HQSLSR, because the SQSLSR uses the ε-dragging technique to relax the binary labels into continuous real values, which enlarges the distances between different classes and makes the discrimination better.

Figure 3 shows the visualization results of the artificial dataset III, which is sampled from two parabolas. Note that our HQSLSR and SQSLSR can obtain parabolic-type regression curves while LSR and DLSR can only obtain straight regression lines, so our methods are more suitable for this nonlinearly separable dataset.

The results of the artificial dataset IV are shown in Figure 4. The nonlinearly separable dataset IV is obtained by sampling from two concentric circles. Obviously, our HQSLSR and SQSLSR have higher accuracy for this classification task, as shown in Figure 4. However, from the first two subfigures, it is not difficult to find that samples of these two classes are far away from their respective regression curves, resulting in poor results of LSR and DLSR. Note that $f_{1}\left(x\right)=0$ and $f_{2}\left(x\right)=1$ coincide and lie at the center of the concentric circles, which are not easy to observe. Thus we only display $f_{1}\left(x\right)=0.1$ and $f_{2}\left(x\right)=0.9$, as shown in last two subfigures.

We conducted experiments on the artificial dataset V to investigate the influence of the ε-dragging technique. The dataset consists of 60 samples from three classes, with 20 samples from each class arranged in three groups: left, middle, and right. By solving the optimization problems of HQSLSR (Equation 15) and SQSLSR (Equation 23) on dataset V, we obtained the corresponding regression labels $\tilde{f}\left(x\right)={\left({\tilde{f}}_{1}\left(x\right),{\tilde{f}}_{2}\left(x\right),{\tilde{f}}_{3}\left(x\right)\right)}^{T}$ and $f\left(x\right)={\left(f_{1}\left(x\right),f_{2}\left(x\right),f_{3}\left(x\right)\right)}^{T}$, where ${\tilde{f}}_{k}\left(x\right),f_{k}\left(x\right),k=1,2,3$ represent the three regression functions solved by HQSLSR and SQSLSR, respectively. The difference caused by the ε-dragging technique is represented by $D=(f\left(x\right)-\tilde{f}\left(x\right))$, which includes three components related to the corresponding three classes. Figure 5 illustrates the relationship between the index of training samples and the three components of the difference D.

According to the results presented in Figure 5b, the first component of the difference matrix D exhibits positive values for the first 20 samples, while negative values are observed for the last 40 samples. This observation suggests that the introduction of the ε-dragging technique has effectively increased the gap in the first component of the difference matrix D between the first class and the remaining classes. Additionally, Figure 5c,d demonstrate that the second and third components of the difference matrix D highlight the second and third classes of samples, respectively. Therefore, the ε-dragging technique has successfully enlarged the differences in regression labels among samples from different classes, thereby enhancing the robustness of the model.

Based on the experimental results presented above, it can be concluded that the regression curve $f_{k}\left(x\right)=1,k=1,2,\cdots ,K$ should be close to the samples from the *k*-th class while being distant from the samples of other classes. The *K* pairs of regression curves can be modeled as arbitrary quadratic surfaces in the plane. This approach enables HQSLSR and its softened version (SQSLSR) to achieve higher accuracy. SQSLSR utilizes the ε-dragging technique to relax the labels, which forces the regression labels of different classes to move in opposite directions, thereby increasing the distances between classes. Consequently, SQSLSR exhibits better discriminative ability than HQSLSR.

### 5.2. Experimental Results on Benchmark Datasets

In order to validate the performances of our HQSLSR and SQSLSR, we compare them with linear methods LSR, DLSR, LDA, SVM-L, LRDLSR, WCSDLSR, and nonlinear methods QSSVM, SVM-R, KRR-R, KRR-P, and reg-LSDWPTSVM. These methods are implemented on 16 UCI benchmark datasets. Numerical results are obtained by repeating five-fold cross-validation five times, including average accuracy (Acc), standard deviation (Std), and computing time (Time). The best results are highlighted in boldface. Lastly, we also calculated the sensitivity and specificity of each method on six datasets to further evaluate their classification performances. Table 1 summarizes the basic information about the 16 UCI benchmark datasets, which are taken from the website https://archive.ics.uci.edu/ml/index.php (the above datasets accessed on 18 August 2021).

In Table 2, we show the experimental results of the above 13 methods on the 16 benchmark datasets. It is obvious from Table 2 that our HQSLSR and SQSLSR outperform linear methods LSR, LDA, DLSR, LRDLSR, WCSDLSR, and SVM-L in terms of classification accuracy on almost all datasets. Moreover, the accuracy of our HQSLSR and SQSLSR are similar to other nonlinear classification methods: SVM-R, SVM-P, KRR-R, KRR-P, QSSVM, and reg-LSDWPTSVM. Note that our SQSLSR has the highest classification accuracy on most datasets. In addition, in terms of computation time, our methods not only have less time cost than the compared nonlinear methods, but also have a narrow gap with the fastest linear method LSR. In general, our HQSLSR and SQSLSR can achieve higher accuracy without increasing the time cost too much, and the generalization ability of SQSLSR in particular is better.

To further evaluate the classification performances of these 13 methods, we show the specificity and sensitivity of the 13 methods on the datasets in Table 3. It can be seen from Table 3, our HQSLSR and SQSLSR perform well in terms of specificity and sensitivity on most of the benchmark datasets.

### 5.3. Convergence Analysis

In this subsection, we experimentally validate the convergence of Algorithm 1. As shown in Figure 6, the value of the objective function monotonically decreases with the increasing number of iterations in six benchmark datasets. Moreover, our SQSLSR converges within five steps on most of the datasets, which indicates Algorithm 1 converges quickly.

### 5.4. Statistical Analysis

In this subsection, we use the Friedman test [31] and the Neymani test [32] to further illustrate the differences between our two methods and other methods.

First, we carry out the Friedman test, where the original hypothesis is that all methods have the same classification accuracy and computation time. We ranked these 13 methods based on their accuracy and computation time on the 16 benchmark datasets and presented the average rank $r_{i}\;\left(i=1,2,\cdots ,13\right)$ for each algorithm in Table 4 and Table 5. Let *N* and *s* denote the number of datasets and algorithms, respectively. The relevant statistics are obtained by
(34)$$\begin{array}{cc} \; & \tau_{\chi^{2}}=\frac{12N}{s(s+1)}\left(\sum_{i}^{s}r_{i}^{2}-\frac{s{\left(s+1\right)}^{2}}{4}\right), \end{array}$$
(35)$$\begin{array}{cc} \; & \tau_{F}=\frac{\left(N-1\right)\tau_{\chi^{2}}}{N\left(s-1\right)-\tau_{\chi^{2}}}, \end{array}$$
where $\tau_{F}$ follows an *F*-distribution with degrees of freedom s−1 and (s−1)(N−1). According to Equation (Equation 35), we obtain two Friedman statistics $\tau_{F}$, which are =12.6243 and 109.9785, and the critical value corresponding to α=0.05 is $F_{\alpha }=1.8063$. Since $\tau_{F}>F_{\alpha }$, we reject the original hypothesis.

Rejection of the original hypothesis suggests that our HQSLSR, SQSLSR, and other methods perform differently in terms of accuracy and computation time. To further distinguish these methods in terms of classification accuracy and computation time, a Nemenyi test is further adopted, and the critical difference is calculated with the following equation:(36)$$\begin{array}{c} CD=q_{\alpha }\sqrt{\frac{s(s+1)}{6N}}, \end{array}$$
when α=0.05, $q_{\alpha }=3.313$, we obtain CD=4.5616 by Equation (Equation 36).

Figure 7 and Figure 8 visually display the results of the Friedman test and the Nemenyi post hoc test. The average rank of each method is marked along the axis. Groups of methods that are not significantly different are connected by red lines.

On the one hand, our methods HQSLSR and SQSLSR are not very different from SVM-R, KRR-R, and KRR-P and are significantly better than LSR, DLSR, LDA, SVM-L, and QSSVM in terms of classification accuracy. On the other hand, our methods HQSLSR and SQSLSR are not very different from LSR, DLSR, and LDA and are significantly better than WCSDLSR, KRR-R, KRR-P, SVM-L, reg-LSDWPTSVM, SVM-R, and QSSVM in terms of computation time. In general, our HQSLSR and SQSLSR can achieve higher accuracy while maintaining relatively small computation time.

## 6. Conclusions

In this paper, utilizing the kernel-free trick and ε-dragging technique, we propose two classifiers, HQSLSR and its softened version (SQSLSR). On the one hand, the quadratic surface kernel-free trick is introduced, which avoids the difficulty of selecting the appropriate kernel functions and corresponding parameters while maintaining good interpretability. On the other hand, utilizing the ε-dragging technique makes the labels more flexible and enhances the generalization ability of SQSLSR. Our HQSLSR can be solved directly, while SQSLSR is solved by an alternating iteration algorithm which we designed. Additionally, the computational complexity, convergence analysis, and interpretability of our methods are also addressed. The experimental results on artificial and benchmark datasets confirm the feasibility and effectiveness of our proposed methods.

In future work, we aim to address several challenges to extend the HQSLSR and SQSLSR models. Specifically, we plan to simplify the quadratic surface to enable our approaches to process high-dimensional data, such as image data. Moreover, we intend to incorporate suitable sparse regularization terms to achieve feature selection.

## Figures and Tables

**Figure 1 entropy-25-01103-f001:**
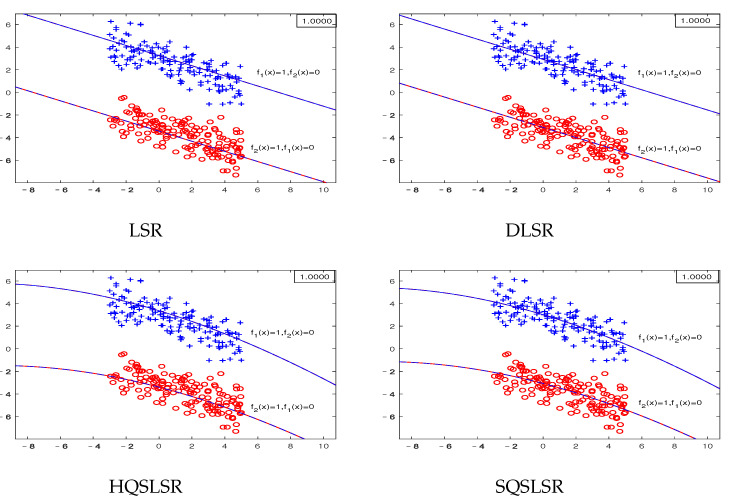
Classification results of the artificial dataset I.

**Figure 2 entropy-25-01103-f002:**
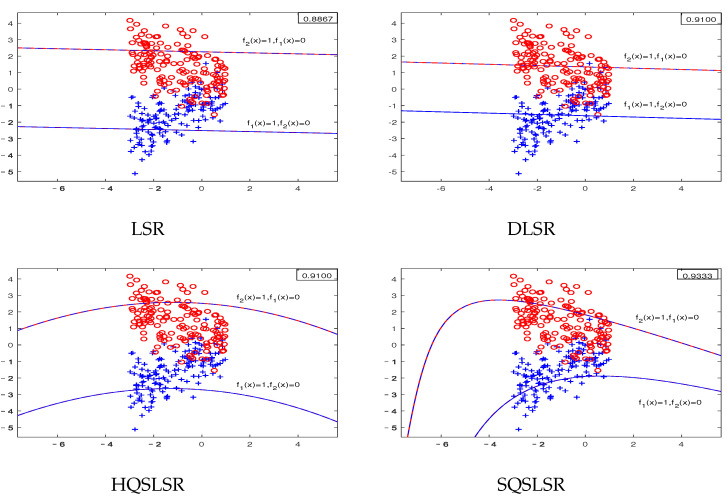
Classification results of the artificial dataset II.

**Figure 3 entropy-25-01103-f003:**
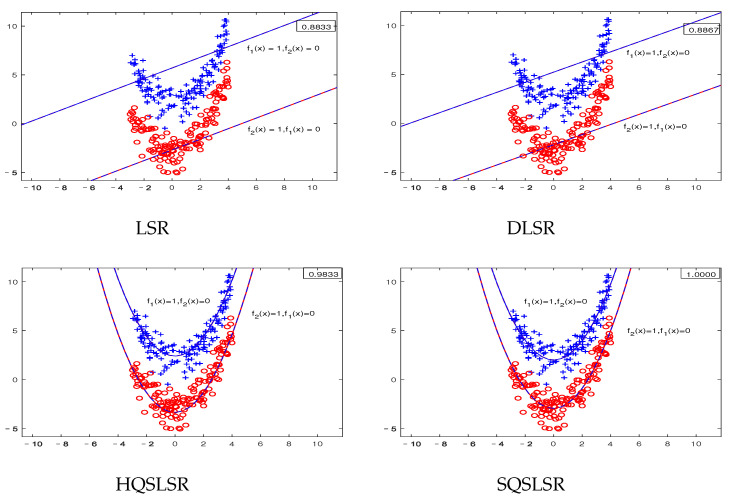
Classification results of the artificial dataset III.

**Figure 4 entropy-25-01103-f004:**
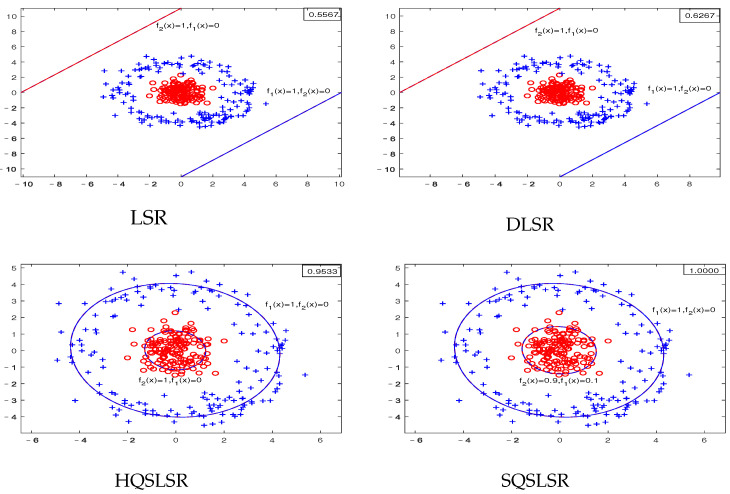
Classification results of the artificial dataset IV.

**Figure 5 entropy-25-01103-f005:**
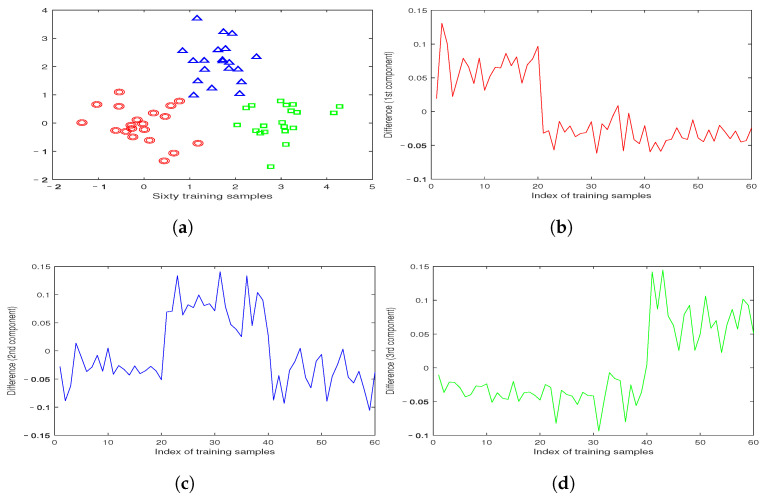
Training samples and the differences caused by ε-dragging technique: (**a**) sixty training samples in three classes; (**b**) the first component of the difference D; (**c**) the second component of the difference D; and (**d**) the third component of the difference D.

**Figure 6 entropy-25-01103-f006:**
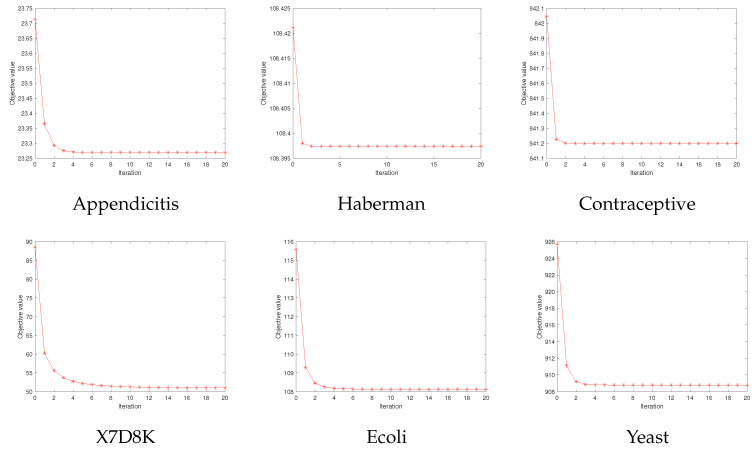
Convergence of SQSLSR.

**Figure 7 entropy-25-01103-f007:**
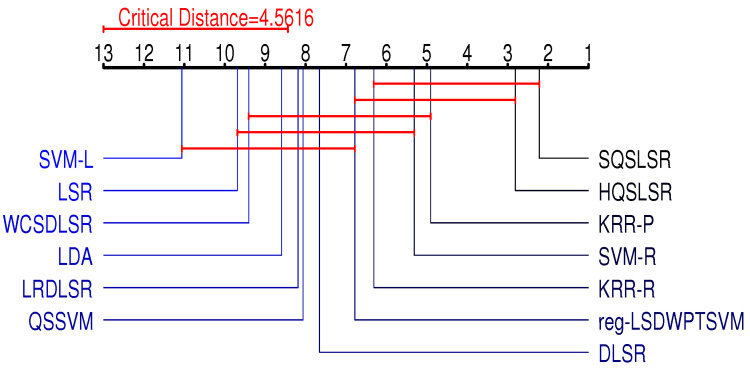
Friedman test and Nemenyi post hoc test of accuracy.

**Figure 8 entropy-25-01103-f008:**
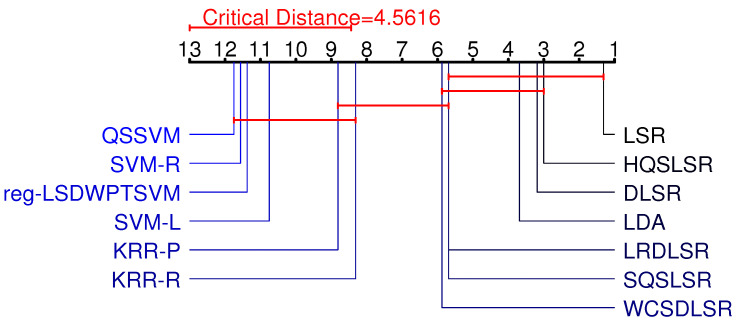
Friedman test and the Nemenyi post hoc test of computation time.

**Table 1 entropy-25-01103-t001:** Basic information of benchmark datasets.

Datasets	Samples	Attributes	Class
Haberman	306	3	2
Appendicitis	106	7	2
Monk-2	432	6	2
Breast	277	9	2
Seeds	210	7	3
Iris	150	4	3
Contraceptive	1473	9	3
Balance	625	4	3
Vehicle	846	18	4
X8D5K	1000	8	5
Vowel	990	13	6
Ecoli	366	7	6
Segmentationation	2310	19	7
Zoo	101	16	7
Yeast	1484	8	10
Led7digit	500	7	11

**Table 2 entropy-25-01103-t002:** Classification results on the 16 benchmark datasets.

		LSR	DLSR	SVM-L	SVM-R	QSSVM	LDA	KRR-R	KRR-P	LRDLSR	WCSDLSR	reg-LSDWPTSVM	HQSLSR	SQSLSR
Haberman	Acc±Std	0.7049±0.0345	0.7377±0.0000	0.7091±0.0383	0.7223±0.0345	0.7158±0.0390	0.6699±0.0490	0.7148±0.0206	0.7411±0.0304	0.7129±0.0135	0.7418±0.0220	0.7158±0,0302	0.7443±0.0245	0.7639±0.0080
	Time (s)	0.0004	**0.0030**	1.0636	1.1224	0.9023	0.0016	0.2093	0.2407	0.0685	0.0086	0.0369	0.0044	0.0048
Monk-2	Acc±Std	0.7763±0.0131	0.7879±0.0135	0.8057±0.0316	0.9954±0.0148	0.9839±0.0213	0.7901±0.0104	0.9424±0.0026	0.9554±0.0001	0.7970±0.0396	0.7546±0.0266	0.9930±0.0104	0.9767±0.0001	0.9770±0.0001
	Time (s)	**0.0008**	0.0030	1.4425	2.4677	1.8716	0.0716	0.4212	0.4564	0.0390	0.0184	0.7327	0.0082	0.0102
Appendicitis	Acc±Std	0.8127±0.0000	0.8286±0.0380	0.8121±0.0638	0.8965±0.0125	0.8485±0.0825	0.6892±0.0534	0.8000±0.0222	0.8667±0.0356	0.8200±0.0213	0.8108±0.0493	0.8675±0409	0.9048±0.0052	0.9143±0.0233
	Time (s)	**0.0010**	0.0032	0.1221	0.1271	0.1310	0.0724	0.0380	0.0119	0.0405	0.0256	1.1540	0.0044	0.0044
Breast	Acc±Std	0.7110±0.0139	0.7201±0.0021	0.7255±0.0410	0.7440±0.0432	0.6571±0.0573	0.6785±0.0418	0.7645±0.0230	0.7174±0.0244	0.7390±0.0532	0.6819±0.0632	0.0.6706±0577	0.7646±0.0182	0.7681±0.0177
	Time (s)	**0.0009**	0.0032	1.0349	0.8887	0.9383	0.0048	0.1807	0.1891	0.0389	0.0077	6.7008	0.0080	0.0086
Seeds	Acc±Std	0.9429±0.0117	0.9619±0.0190	0.8667±0.0614	0.9286±0.0261	0.9143±0.0190	0.9667±0.0117	0.9571±0.0178	0.9762±0.0150	0.9762±0.0337	0.0.9524±0.0238	0.0.9581±0.0.0433	0.9810±0.0095	0.9857±0.0117
	Time (s)	**0.0027**	0.0070	0.6734	0.9577	0.7920	0.0067	0.1166	0.1360	0.0393	0.0096	1.7335	0.0058	0.0474
Iris	Acc±Std	0.8333±0.0365	0.8400±0.0249	0.7200±0.0691	0.9667±0.0298	0.9333±0.0333	0.9467±0.0163	0.9533±0.0339	0.9662±0.0163	0.8333±0.0572	0.8133±0.0298	0.9600±0.0149	0.9733±0.0249	0.9667±0.0030
	Time (s)	**0.0040**	0.0028	0.3334	0.4720	0.2308	0.0042	0.0590	0.0640	0.0400	0.0053	0.1385	0.0032	0.0032
Contraceptive	Acc±Std	0.5031±0.0172	0.5088±0.0216	0.3508±0.0246	0.5479±0.0153	0.4379±0.0425	0.5112±0.0482	0.5427±0.0185	0.5417±0.0230	0.4939±0.0268	0.4996±0.0199	0.4773±0.0321	0.5475±0.0112	0.5448±0.0171
	Time (s)	**0.0033**	0.0340	50.5654	49.5618	152.4766	0.0197	5.6963	6.4789	0.0836	1.0946	39.7778	0.0478	0.4666
Balance	Acc±Std	0.8592±0.0099	0.8609±0.0027	0.8384±0.0391	0.9002±0.0274	0.9440±0.0236	0.6880±0.0209	0.9121±0.0073	0.9105±0.0078	0.8739±0.0146	0.0.8824±0.0409	0.0.9056±0.0215	0.9153±0.0063	0.9162±0.0062
	Time (s)	**0.0022**	0.0100	0.8838	6.8447	1.8852	0.0050	1.0122	1.0689	0.0703	0.1482	0.1496	0.0122	0.6072
X8D5K	Acc±Std	1.0000±0.0000	1.0000±0.0000	0.8750±0.0040	1.0000±0.0000	0.9860±0.0020	1.0000±0.0000	1.0000±0.0000	1.0000±0.0000	1.0000±0.0000	1.0000±0.0000	1.0000±0.0000	1.0000±0.0000	1.0000±0.0000
	Time (s)	0.0134	**0.0023**	17.1786	19.3314	41.5740	0.0617	3.4147	3.7119	0.1361	0.4812	27.1015	0.0277	0.1834
Vehicle	Acc±Std	0.7521±0.0335	0.7686±0.0238	0.6399±0.0631	0.6661±0.0374	0.7694±0.0305	0.7694±0.0375	0.7675±0.0319	0.8287±0.0328	0.7637±0.0439	0.7471±0.79	0.7494±0.0148	0.8229±0.0207	0.8321±0.0066
	Time (s)	**0.0025**	0.0356	21.2842	25.7992	414.1790	0.0737	2.4283	1.9887	0.0976	0.4068	4872.9805	0.0810	0.1314
Zoo	Acc±Std	0.9328±0.0249	0.9399±0.0200	0.8910±0.0306	0.9210±0.0406	0.8819±0.0481	0.8654±0.0250	0.9299±0.0302	0.9474±0.0008	0.9437±0.0598	0.0.9210±0.0266	0.0.9505±0.0354	0.9527±0.0028	0.9600±0.0020
	Time (s)	0.0118	0.0179	0.6321	0.2503	2.3420	0.0358	0.1827	0.2904	0.0840	0.0161	3486.3605	**0.0072**	0.0540
Yeast	Acc±Std	0.5508±0.0161	0.5684±0.0107	0.5162±0.0373	0.6004±0.0165	0.5596±0.0055	0.5045±0.0138	0.5926±0.0202	0.6007±0.0185	0.5354±0.0210	0.0.5451±0.0266	0.5445±0.0145	0.6097±0.0224	0.6154±0.0183
	Time (s)	**0.0058**	1.3578	145.6627	158.0168	109.6964	0.0837	12.8849	27.3849	0.2602	2.2958	132.7344	0.0452	1.6300
Ecoli	Acc±Std	0.7136±0.0135	0.7482±0.0240	0.7469±0.0418	0.8900±0.0341	0.8007±0.0271	0.8544±0.0254	0.7317±0.0200	0.8928±0.0146	0.7977±0.0265	0.0.8303±0.0313	0.0.8720±0.0523	0.8927±0.0254	0.8751±0.0172
	Time (s)	**0.0020**	0.0518	4.3479	5.6260	5.8922	0.0037	2.2895	1.4722	0.1105	0.1738	8.5413	0.0088	0.0594
Led7digit	Acc±Std	0.7177±0.0261	0.7349±0.0274	0.5420±0.0105	0.6820±0.0000	0.6660±0.0543	0.7420±0.0264	0.7331±0.0374	0.7246±0.0147	0.7138±0.0497	0.7040±0.0241	0.0.6960±0.0456	0.7407±0.0367	0.7412±0.0236
	Time (s)	**0.0031**	0.4414	14.8471	81.7604	29.9238	0.0072	1.3508	1.4445	0.1476	0.3050	0.25.4077	0.0116	0.4652
Vowel	Acc±Std	0.4335±0.0201	0.4354±0.0312	0.4101±0.0215	0.9848±0.0090	0.8192±0.0200	0.5722±0.0232	0.9939±0.0059	0.8131±0.0209	0.4647±0.0369	0.3979±0.0438	0.9556±0.0131	0.8202±0.0336	0.8667±0.0174
	Time (s)	**0.0039**	0.2780	74.6948	81.7602	485.9184	0.1485	5.7673	11.1241	0.3047	2.0553	1044.9018	0.0434	3.1902
Segmentation	Acc±Std	0.8403±0.0025	0.8403±0.0096	0.9307±0.0071	0.9476±0.0146	0.9392±0.0114	0.9100±0.0118	0.9420±0.0068	0.8952±0.0050	0.8666±0.0112	0.8429±0.0309	0.9221±0.0592	0.9429±0.0060	0.9483±0.0000
	Time (s)	**0.0060**	0.4242	299.6623	294.5338	3053.9000	0.3877	23.1303	20.1449	0.2635	6.6594	8310.9828	0.2028	3.9048

**Table 3 entropy-25-01103-t003:** Specificity and sensitivity results of each method.

Dataset	Sensitivity						Specificity					
	Appendicitis	Haberman	Contraceptive	X8D5K	Ecoli	Yeast	Appendicitis	Haberman	Contraceptive	X8D5K	Ecoli	Yeast
LSR	0.2273	0.2143	0.4740	**1.0000**	0.7247	0.3986	0.9375	0.9551	0.7434	**1.0000**	0.9709	0.9389
DLSR	0.4400	0.2250	0.4788	**1.0000**	0.7167	0.3814	0.9647	0.9511	0.7422	**1.0000**	0.9704	0.9405
SVM(line)	0.4000	0.1875	0.4016	0.9910	0.8559	0.4677	0.9412	0.9200	0.6958	0.9977	0.9667	0.9357
SVM(rbf)	0.5000	0.3058	0.4755	**1.0000**	0.8476	0.5533	0.9294	0.8444	0.7403	**1.0000**	0.9655	0.9424
QSSVM	0.5142	0.2070	0.3530	0.9960	0.7014	0.4062	0.9412	0.9467	0.7424	**1.0000**	0.9659	0.9361
LDA	0.5633	**0.5214**	0.4871	**1.0000**	0.8223	0.5556	0.6592	0.7236	0.7584	**1.0000**	0.9609	0.9398
KRR-R	0.4521	0.2222	**0.5280**	**1.0000**	0.7139	0.5552	0.9306	0.9387	0.7626	**1.0000**	0.9705	**0.9467**
KRR-P	0.4948	0.3000	0.5234	**1.0000**	0.8536	0.5367	0.9640	0.9376	0.7635	**1.0000**	0.9717	0.9339
LRDLSR	0.400	0.2250	0.6128	**1.0000**	0.5317	0.3036	0.9333	0.9504	0.7439	**1.0000**	0.9662	0.9354
WCSDLSR	0.3333	0.3684	0.4653	**1.0000**	0.7854	0.3269	0.9444	0.9446	0.7370	**1.0000**	0.9630	0.9380
reg-DWPDSVM	0.4867	0.4111	0.4730	**1.0000**	0.8540	0.5248	0.9422	0.9149	0.7300	**1.0000**	0.9716	0.94511
HQSLSR	0.5700	0.3875	0.5249	**1.0000**	8581	0.5575	0.9647	0.9467	0.7671	**1.0000**	0.9765	0.9465
SQSLSR	**0.6824**	0.3176	0.5226	**1.0000**	**0.8647**	**0.5629**	**0.9667**	**0.9511**	**0.7795**	**1.0000**	**0.9797**	0.9460

**Table 4 entropy-25-01103-t004:** Ranks of accuracy.

Datasets	LSR	DLSR	SVM-L	SVM-R	QSVM	LDA	KRR-R	KRR-P	LRDLSR	WCSDLSR	reg-LSDWPTSVM	HQSLSR	SQSLSR
Haberman	12	5	11	6	7.5	13	9	4	10	3	7.5	2	1
Monk-2	12	11	8	1	3	10	7	6	9	13	2	5	4
Appendicitis	9	7	10	3	6	13	12	5	8	11	4	2	1
Breast	9	7	6	4	13	11	3	8	5	10	12	2	1
Seeds	10	6	13	11	12	5	8	3.5	3.5	9	7	2	1
Iris	10.5	9	13	2.5	8	7	6	4	10.5	12	5	1	2.5
Contraceptive	8	7	13	1	12	6	4	5	10	9	11	2	3
Balance	11	10	12	7	1	3	4	5	9	8	6	3	2
X8D5K	6	6	13	6	12	6	6	6	6	6	6	6	6
Vehicle	9	6	13	12	4.5	4.5	7	2	8	11	10	3	1
Zoo	7	6	11	9.5	12	13	8	4	5	9.5	3	2	1
Yeast	8	6	12	4	7	13	5	3	11	9	10	2	1
Ecoli	13	10	11	3	8	6	12	1	9	7	5	2	4
Led7digit	7	4	13	11	12	1	5	6	8	9	10	3	2
Vowel	11	10	12	2	6	8	1	7	9	13	3	5	4
Segmentation	12.5	12.5	6	2	5	8	4	9	10	11	7	3	1
Average ranks	9.6875	7.65625	11.0625	5.3125	8.0625	8.59375	6.3125	4.9062	8.1875	9.40625	6.78125	2.8125	2.21875

**Table 5 entropy-25-01103-t005:** Ranks of computation time.

Datasets	LSR	DLSR	SVM-L	SVM-R	QSVM	LDA	KRR-R	KRR-P	LRDLSR	WCSDLSR	reg-LSDWPTSVM	HQSLSR	SQSLSR
Haberman	1	3	12	13	11	2	9	10	8	6	7	4	5
Monk-2	1	2	11	13	12	7	8	9	6	5	10	3	4
Appendicitis	1	2	10	11	12	9	7	5	8	6	13	3.5	3.5
Breast	1	2	12	10	11	3	8	9	7	4	13	5	6
Seeds	1	4	10	12	11	3	8	9	6	5	13	2	7
Iris	4	1	12	13	11	5	8	9	7	6	10	2.5	2.5
Contraceptive	1	3	12	11	13	2	8	9	5	7	10	4	6
Balance	1	3	9	13	12	2	10	11	5	6	7	4	8
X8D5K	2	1	10	11	13	4	8	9	5	7	12	3	6
Vehicle	1	2	10	11	12	3	9	8	5	7	13	4	6
Zoo	2	4	11	9	12	5	8	10	7	3	13	1	6
Yeast	1	5	12	13	10	3	8	9	4	7	11	2	6
Ecoli	1	4	10	11	12	2	9	8	6	7	13	3	5
Led7digit	1	6	10	13	12	2	8	9	4	5	11	3	7
Vowel	1	4	10	11	12	3	8	9	5	6	13	2	7
Segmentation	1	5	11	10	12	4	9	8	3	7	13	2	6
Average ranks	1.3125	3.1875	10.7500	11.5625	11.7500	3.6875	8.3125	8.8125	5.6875	5.8750	11.3750	3	5.6875

## Data Availability

All of the benchmark datasets used in our numerical experiments are from the UCI Machine Learning Repository, which are available at https://archive.ics.uci.edu/ml/index.php (the above datasets accessed on 18 August 2021).

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
