# Peer review of "Kernel-Free Quadratic Surface Regression for Multi-Class Classification"

_entropy, 2023, doi:10.3390/e25071103_

Round 1

Reviewer 1 Report

In this manuscript, two quadratic surface least squares regression models are proposed for multi-class classification problems. The computational results are not bad. But this manuscript is not well-written. I have some concerns as the following before reconsidering this manuscript:

1. Why propose an alteration iteration algorithm to solve the SQSLSR, which is a simple convex optimization problem with the simple bound of some variables? The well-known solvers can be used to solve this model efficiently and accurately. Please specific the detailed needs and advantages of designing this algorithm.

2. In the literature review of this manuscript, the advantages and disadvantages of some related methods in the references are not talked about in details so that the logic between the references is not very clear. Please introduce them in more details. Moreover, some recent related references about kernel-free SVM or SVR models (including the papers about Unsupervised quadratic surface SVM with application to credit risk assessment, A kernel-free double well potential SVM with applications and so forth) are missed in the literature review.

3. There are few state-of-the-art multi-classification models in the compared models in the numerical experiments of this manuscript. Why not compare the proposed models with some recently-developed regression-based or kernel-free SVM-based multi-classification models (including the methods proposed in the reference [6], [11] and [23] in this manuscript)? These models are highly related to the proposed models in this manuscript.

4. Although the procedure of selecting a kernel and its related parameters is avoided,  the efficiency of proposed model is also important. Please also include the computational time of all tested models for comparisons in the numerical experiments.

5. The titles of some sections or subsections should be revised for more specific and suitable ones.

6. The styles of references are not consistent. And the references [22] and [26] are the same.

There are many typos and informal language usage. Please greatly improve the readability of this manuscript by carefully checking the whole manuscript.

Reviewer 2 Report

The authors have suggested a new optimization framework for multi-class segmentation problem. The presentation of the proposed work is easily tracked however the comparison of the computational burden will be an important factor for highlighting the advantage of the proposed work. I kindly suggest adding the timeline for the training of the models. 

Round 2

Reviewer 1 Report

Please improve the readability of this manuscript by carefully correct the informal language usage or typos in the whole manuscript

Please improve the readability of this manuscript by carefully correct the informal language usage or typos in the whole manuscript

Reviewer 2 Report

Thank you for revising the article with the reviewers' comments. I have only minor comments for the authors. Format of the citation should be the same, especially citing other research using authors, "last_name et al. [XX]". Alos, please change the title of section 4 to "Discussion"
